# Simple Universal Whole-Organ Resin-Embedding Protocol for Display of Anatomical Structures

**DOI:** 10.3390/biomedicines11051433

**Published:** 2023-05-12

**Authors:** Ionica Pirici, Liliana Cercelaru, Diana Iulia Stanca, Andrei Osman, Lorena Sas, Daniel Pirici, Ion Mindrila

**Affiliations:** 1Department of Human Anatomy, University of Medicine and Pharmacy of Craiova, 200349 Craiova, Romania; ionica.pirici@umfcv.ro (I.P.); andrei.osman@umfcv.ro (A.O.); lorena.sas@umfcv.ro (L.S.); ion.mindrila@umfcv.ro (I.M.); 2Department of Neurology, University of Medicine and Pharmacy of Craiova, 200349 Craiova, Romania; liliana.cercelaru@umfcv.ro (L.C.); iulia.stanca@umfcv.ro (D.I.S.); 3Department of Histology, University of Medicine and Pharmacy of Craiova, 200349 Craiova, Romania

**Keywords:** whole-organ plastic embedding, acrylic resin, anatomy specimens, display specimens

## Abstract

Whole-organ plastic resin casting is a very useful method for preserving rare pathological specimens for forensic/anatomical studies and for teaching/research purposes. Many techniques have been proposed over time, but most of them use special non-commercially available resin mixtures, lengthy protocols, and are overall not easily implemented in any anatomy/pathology department that might need such a procedure for rapid organ preservation. Here, we utilized anatomical sections of the human brain, heart, kidneys, spleen, large intestine, and lungs from on-display organs that were fixed for more than 1 year in 10% neutral-buffered formalin and from a freshly processed cadaver for teaching purposes in our Human Anatomy Department, and we optimized a fast-processing protocol without the use of any clearing agents, which yields solid, clear, cylindrical resin casting blocks. The resulting protocol, which takes no longer than 4 days, proves that at least three commonly used epoxy resins from hobby shops can be utilized without any restrictions, and the use of resin or glycerin vacuum-forced impregnation even offers two choices of intrinsic contrast, depending on the nature of the preparation. A number of innovations have been included here and compared to existing publications, such as the use of a system of permanent fixation plexiglas rods that maintain the organ in the desired position and become invisible in the final block, the use of UVC sterilization of the tissue to ensure a long shelf life of the block, and the utilization of cheap cylindrical polypropylene food containers as casting molds. Altogether, we present a simple resin-embedding protocol that can be made available to any department/institution without the need for expensive materials and specially trained personnel.

## 1. Introduction

The preservation of gross anatomical specimens represents an important source of material for forensic macroscopy evidence analysis, as well as anatomy teaching and research purposes. Therefore, efforts are being made to replace old-fashioned open-jar display preparations with epoxy resins/liquid plexiglas-embedded specimens that are more easily handled, less toxic, and ideally, more resistant over time.

Many early methods of tissue processing for impregnation in different rigid materials have been proposed, with an essential advancement in the field being the introduction of plastination techniques starting with Gunther von Hagens in 1977 [1,2,3]. The main principle behind the process of plastination is the removal of tissue fluid and its replacement with a curable polymer resin that either infiltrates and maintains the shape of the respective organ, or it completely submerges the structure in a solid block of resin. The technique was later improved and developed by different authors [4,5,6]. They tried to obtain structures with a maximum degree of transparency that were sufficiently resistant over time, and most importantly, to preserve morphological and structural features to be used for didactic or research purposes. Different epoxy resins and silicones have been proposed over time; however, all these procedures require lengthy vacuum-forced dehydration steps, sometimes in freezing conditions, with complex timing and even customized resin formulations—procedures that might not be easily applicable in anatomy/pathology departments with teaching personnel who are not specialized in these applications and have limited instruments.

The purpose of the present study was to validate a simple and rapid resin-casting technique that does not use organic solvents, so as to preserve the fatty tissue as much as possible, as well as to utilize simple and widely available plastic resins and ancillaries. We thus tested three of the most common resins sold in local hobby shops that are used for plastic moldings and show that all three can provide good results, with two of them withstanding long UVC exposures as an aging test. Moreover, we aimed to bring some innovations to the technique, such as a method of floating the anatomical sample into the fluid resin without casting multiple layers, pre-inclusion UVC exposure for sterilization, and the use of cheap, commercially available polypropylene cylindrical containers as a better alternative to custom-made plexiglas boxes. This technique will offer a simple option for producing high-quality anatomical preparations for both teaching display use and morphological research studies.

## 2. Materials and Methods

### 2.1. Specimens and Tissue Fixation

The aim of this study was to compile and test a simple and universal plastic resin-embedding method for anatomical specimens. Therefore, we wanted the technique to be applicable both to specimens displayed in formalin for longer periods of time, or processed according to more controlled fixation periods, and to tissues with different degrees of fatty elements. Representative anatomical sections/blocks were obtained from human brain hemispheres (*n* = 5), heart (*n* = 4), kidney (*n* = 3), spleen (*n* = 2), 4 cm-long transverse large intestine segments with mesentery (*n* = 2), and lung lobes prepared as 1 cm-thick slices (*n* = 2). These were taken from either on-display organs fixed for more than 1 year in 10% neutral-buffered formalin (NBF) or from a freshly processed cadaver for teaching purposes in the Human Anatomy Department of the University of Medicine and Pharmacy of Craiova. The fresh tissue was flushed of any content and thrombi, trimmed, and then fixed in NBF for one month before being processed for resin embedding. Representative anatomical sections were obtained from tissues with different degrees of fat components in order to evaluate the capacity of the method to preserve fatty elements in the structure of different organs.

Furthermore, a freshly fixed lung lobe was prepared for selective staining prior to resin embedding. Briefly, after the formalin was washed away, available bronchi were cleansed by cannulation with distilled water at a pressure of 20 cm H_2_O, and then the parenchyma was gently compressed to drain it out. This was repeated until the draining water was clean. Then, a segmental bronchus was slowly injected with a red water-soluble dye (commercial washable paint), with a pressure of 20 cm H_2_O. A 1 cm-thick slice was then cut from the parenchyma and further processed for resin embedding utilizing glycerol vacuum impregnation, as described below.

### 2.2. Tissue Processing for Resin Embedding

Once the tissue fragments were fixed, they were washed in running tap water overnight (12 h, room temperature) in order to remove, as much as possible, the fixative and to prevent artefactual precipitations, as well as toxic vapor formation.

In order to remove the water, the tissues were next dehydrated in a graded ethanol series, ranging from 70% to two absolute ethanol incubation steps (Figure 1). For consistency, all water and ethanol processing were performed by programming the timings in a rotary tissue processor available in the Histology Department within our University (STP 120, Thermo Fischer Scientific, Waltham, MA, USA). Next, since we intended to alter the morphology as little as possible, we did not continue with a solvent-clearing method that would have dissolved the fatty elements; instead, we brought the tissue either into inactivated resin or glycerol, a step that offers a protective interface between the tissue and the activated resin. In order to impregnate the tissue, this step (glycerol or inactivated resin) was performed under vacuum (−1 bar) in a simple resin-degassing vacuum setup for 1 h (generic single stage 85 L/min vacuum pump up to 5 Pa, a 19 L steel vacuum chamber with a 2 cm-thick transparent acrylic lid, valves, oil manometer, and vacuum hose) (Appendix A). Next, the organs were tapped of excess resin or glycerol and slowly lowered into the final activated resin in a volumetrically adequate mold.

In order to develop a simple universal technique, we chose some of the most frequently sold bicomponent plastic resins utilized for modeling casting from two online Romanian hobby shops: (i) Plexi Fluid 2.0 (Catalog # FT000006, Prochima S.R.L., Calcinelli di Saltara, Italy). This was recommended by the producer for thick casting without yellowing in time, as well as for its resistance to UV, its lack of exergonic reaction during polymerization, lack of upper limit of the total volumes, and lack of toxic vapors, with a mixing mass ratio of 10/3); (ii) Epoxy Pro Klar (Vosschemie GmbH, Uetersen, Germany), a solvent-free resin that is completely transparent, with a mixing mass ratio of 100/40); and (iii) GTS-Incluziune (Polyesteric resin, Vosschemie GmbH), which is UV stable, needs to be utilized in a well-ventilated place, as it can be irritating and produce stiren fumes, and has a ratio of the hardening component of 0.5% of the final mass. For all the formulas, we utilized the middle-range concentrations of the activator recommended by the producer in order to slow the reaction as much as possible and reduce the oxidation and heating effects.

As these resins have a high density, and the biological tissue would float to the surface of the fluid resin, we did not intend to unnecessarily extend the working protocol for casting/curing multiple layers of resin in order to fix the organs in the desired position, as this might also inadvertently expose the upper layers of the tissue to air and lead to a non-uniform color. Therefore, in order to place the tissue inside the volume of the final resin block with a “floating effect”, we 3D printed a lid in which we inserted thin vertical 4 mm rods of different materials (we tested transparent acrylic plastic and wood) to keep the tissue sunken until the resin was viscous enough to retain the tissue but which would still to allow us to remove the supports (Figure 2). The lid design was made with the Autodesk Tinkercad platform (Autodesk Inc., San Francisco, CA, USA) and printed on a Tiertime UpBoxPlus 3D printer (Tiertime Technology Co. Ltd., Beijing, China) with 1.75 mm polylactic acid (PLA) filament. The stereolithography CAD files (stl.) are available as a Appendix A or can be requested from the authors.

Almost all of the previously published casting techniques utilize plexiglas plates to cast parallelepipedal blocks of resin, which first need to be accurately trimmed, framed in the desired position, and glued in such a way that minute edges should not distort the view of the specimen inside. More importantly, creating 90° corners definitely distorts the view of anatomical details on the surface of the organ, with different thicknesses of the overlaying resin acting like lenses. Therefore, we needed a much more useful and simple solution, and we chose commercially available cylindrical polypropylene food-packing buckets of different volumes, which served as molds for the resin, with their cylindrical conformation offering the same resin thickness all around the organ. We also tested the usefulness of glass cylindrical recipients for this purpose and assessed whether cured resins could be easily extracted from polypropylene/polyvinyl chloride (PVC)/glass molds.

In our protocol, we also utilized a handheld sanding machine with sandpaper disks with granulations of 180, 240, and 1000, plastic polishing cream (Up System M-150, Vosschemie GmbH), and a laser-guided generic non-contact infrared thermometer.

## 3. Results

### 3.1. Overall Processing and Embedding Protocol

All the steps of the protocol are extremely simple to follow and are highly reproducible (Figure 1). After fixation and dehydration, excess ethanol was adsorbed by wrapping the organs in tissue paper. Next, organ sections were suspended in inactivated resin compounds, with or without a glycerol-embedding step, utilizing the 3D-printed disk and 4 mm wood and transparent plexiglas rods to keep the organ “floating” in the middle of the resin cast. Considering the mixing ratio intervals of the two components of the resins, and as recommended by the producers, we always aimed for the middle of the range in order to obtain a balanced polymerization speed and also to avoid heat formation.

For prevention of the formation of small air bubbles on the container’s walls, we observed that it is best to first mix the two parts of the resin in the container (utilizing a spoon until the compound becomes homogenous), degas the mix alone, then utilize the spoon again to remove bubbles attached to the inner walls of the container, and finally, degas the activated resin one more time. Meanwhile, the tissue fragment was wrapped in tissue paper to remove the excess absolute ethanol or glycerin, depending on which of the two protocols was followed (Figure 1). Reasoning that despite fixation, glycerol can be a medium that induces molds inside the tissue in the case of contamination, we also included a 30 min germicidal ultraviolet light band C (UVC quartz lamp) exposure step, just before submerging the samples into the resin. The tissue was next slowly lowered into the resin, taking care that no air pockets would form underneath it, and the 3D-printed support and rods were used to maintain the tissue at the desired depth in the resin. Vacuum degassing should only be applied for direct resin impregnation, as glycerol-impregnated tissue is already vacuumed during the glycerol incubation step. Inspection of the process through the transparent lid of the degassing chamber was essential to ensure the complete degassing of the sample. Vacuum levels were always varied gradually and slowly, utilizing the pump and, respectively, the control valve, considering that violent pressure variations can lead to temperature variations and bubble formation. The jar with the sample was then carefully removed from the vacuum chamber and placed on a shelf with tissue paper on top, in order not to allow dust to settle on the upper layer of the resin. Careful movement and adjustments of the supporting rods can still be performed at this point in order to ensure the desired position and orientation of the organ.

Depending on the type of resin utilized and the mix ratios of the two components, we observed that between 12–24 h after activation, the viscosity of the resin increased enough to allow the organ to remain trapped inside, and thus, by carefully monitoring the temperature, it was possible to slowly remove each individual rod from the resin without leaving visible traces inside. This, however, required relatively close monitoring of the process, as the time frame was not longer than 3–4 h, after which it was difficult to remove the rods without displacing large parts of resin. We monitored the temperature of the resin utilizing an infrared thermometer, and we observed that the temperature was relatively constant throughout the polymerization process (room temperature), except for the curing interval mentioned above, when it increased to approximatively 35–40 °C. Therefore, theoretically, it is possible to conceive a temperature sensor that would start an alarm indicating the best time for removing the supporting rods. However, we observed that transparent plexiglas rods submerged in the resin, with approximatively the same refractive indexes, become invisible, and therefore, in the end, our final optimized protocol used only this type of material for support, and we did not remove them at all from the final block (Figure 3). We also tested the protocol with different resin ratios, and when approaching the maximum ratio of activator/total mass, the polymerization was much faster (taking less than 5 h), the samples became very hot (>90 °C), and the specimen was completely covered with gas bubbles, destroying the display sample (Appendix A).

After 48 h, all the resin blocks could be easily removed from their polypropylene molds, rendering an almost finite aspect to the sample. Utilizing PVC or glass molds, however, made it impossible to extract the resin block, as all three proposed compositions adhered strongly to these last two materials. We also confirmed that after complete polymerization, submerged plexiglas rods were still invisible in the resin, greatly simplifying the orientation process of the tissue.

We also performed an accelerated aging test by exposing the blocks to UVC (20 W lamp, and at approximatively 30 cm) continuously for 5 days. The Epoxy Pro Klar block started to show a “yellow” tint, whereas the other two products maintained a completely unchanged transparent aspect. This UVC resistance can be utilized as a sterilization method against mold and bacteria growth inside the tissue blocks; however, due to the fact that the producers do not list the UVC permeability of their final cured resins, it is best to expose the samples to UVC just prior to inclusion in the activated resin.

After curing, the resin blocks can be trimmed with a saw if necessary, leveled with a sanding machine using sand paper disks with increasing granularity (we utilized 180-, 240-, and 1000-grit disks), and in the end, polished with a soft wool disk and polishing cream. All the surfaces of the resin blocks can thus be leveled and polished to complete transparency, and essentially, all the plexiglas supporting rods can now also be polished until they are barely visible on the surface of the resin. This applies to all three types of resins tested here (Figure 3).

Our initially proposed protocol also contained two main conditions of inclusion: direct resin vacuum impregnation of the sample, or glycerol vacuum impregnation and then resin casting without re-vacuuming (Figure 1). These options were explored in order to see whether exposing the tissue to the oxidizing effect of the curing resin would influence the color of the finite tissue. Indeed, using direct impregnation with the inactivated resin led to more dark-colored tissue compared to glycerol impregnation, with both end results being equally useful, depending on the purpose of the display sample. For example, a glycerol-impregnated sample best retained the natural color of the heart and kidney, and a resin-impregnated brain slice much better revealed the contrast between the white and gray matter (Figure 4).

We also utilized some red/blue PLA filament fragments to illustrate elements of interest in the organ, such as arteries or veins, and these were fully compatible with all the tested resins in that they were not dissolved or deformed during polymerization (Figure 4, Appendix A).

### 3.2. Final Working Protocol

The optimized final protocol is highly reproducible and simple to follow for any personnel, without any special training necessary:Trim and fix the specimen for at least a month;Dehydrate till absolute ethanol (Figure 1);Expose to UVC for 30 min;Forced-vacuum glycerin or inactivated resin impregnation; print the PLA support of the transparent plexiglas rods; prepare colored PLA filament fragments for labeling purposes;Prepare the activated resin in the polypropylene mold using two degassing/mixing cycles to remove all bubbles, cut and trim the length of the transparent plexiglas rods to maintain the organ at the desired depth in the mold;Gently lower the organ into the degassed resin, and then place the PLA lid with the plexiglas rods on top; make adjustments as necessary to level the organ (by retracting/extending the rods through the PLA lid, even organs with irregular surfaces can be maintained horizontally);Allow 48 h for curing;Remove the resin block from the mold, cut the transparent rods as close to the surface of the resin as possible, and polish the upper surface with increasing granulation disks until sanding cream;Print and place a QR code on the cured block; this can allow for the description of that organ in the anatomy/histology database of the institution, to allow students fast access to the detailed organ data (optional);Repair future defects on the surface of the resin block by applying more activated resin and polishing.

From our tests, the best choice of resin was the Plexi Fluid 2.0, which does not release toxic fumes and maintains its perfect transparency after five days of UVC exposure. With respect to the date of submitting this manuscript, our initial blocks were produced 11 months earlier, and no further changes in color were detectable. Although we tested maximum casting volumes of 700 mL, this resin is advertised as having no upper limit for the total working volumes; therefore, larger casts are also possible.

No observable final differences could be noted between 1 month of fixation and BNF archived tissue.

## 4. Discussion

Embedding whole organs or large macroscopic tissue sections in transparent resins for preservation purposes is not new, but it is a field in need of continuous improvement. Faster and simpler processing methods that can allow for more anatomy/pathology/museum departments to perform in-house protocols in order to obtain durable and high-quality display material are also needed.

The first report of using a resin to render tissues transparent is generally attributed to Werner Spalteholz, a German anatomist who described the method in 1914 as a technique of rendering biological tissues transparent by soaking them in a mixture of alcohol and methyl salicylate, with the resulting specimens revealing internal structures with unprecedented clarity, which was a significant advance in the field of anatomy and histology [7]. In 1949, Gough and Wentworth were successful in preparing lungs in real-life volumes by inflating the lung and submerging it in a fixative solution [8]. The lungs were then sliced into thin sections by hand using a knife and impregnated with gelatin of increasing concentrations. Finally, the sections were embedded into a formol–gelatin mixture and cut into thin slices using a large microtome. These slices were then placed on “perspex” sheets, covered with absorbent filter paper, and left to dry. The dried sections adhered to the paper, and the resulting specimens were filed and displayed in a book format [8,9].

Post-mortem dye-injection techniques were historically used to highlight the vascular system. In 1936, a method was used to combine colored dye injection and corrosion of the tissue by fumaric acid, with whole-organ vinyl resin casting in order to visualize canalicular structures as the vasculature, bile ducts, or bronchiolar tree [10,11,12]. Unfortunately, some of the substances that were used caused contact dermatitis and required a lot of time for inclusion (several weeks), and the fluids used to fix and mount the preparations influenced the color and shape of the preserved specimen.

A technique of embedding dry anatomical specimens in organic glass (methyl methacrylate) was published in 1960. Basically, the dried specimens were infiltrated with the pre-prepared monomer under vacuum conditions, then dipped into activated polymer that was poured into an enclosure made of methyl methacrylate plates that became part of the block once the polymerization took place. This technique was most promising due to the fact that it could quickly lead to a solid block of transparent plastic, but it still had many disadvantages, such as the need to distill the polymer at different temperature points and to use custom-sized boxes made of in-house trimmed plastic plates for each specimen, as well as the fact that the proper dehydration of large specimens was not properly addressed [13]. Plastination was invented in 1946 by Romaniak, as a technique of tissue preservation in which water and lipids from biological specimens were gradually replaced by curable polymers that then harden, leading to hard, dry, odorless display specimens that should ideally retain the original shape and color of the respective organs [14]. Romaniak utilized unpolymerized resin, and later on, Von Hagens and others used polymerizing resins to gradually replace intermediary solvents such as acetone and xylene [1,2,15,16,17,18]. Different types of resins have been reported with different results regarding the color and volume of the final display specimens, and the main advantage of the plastination technique is that it allows for the preservation of the complete anatomical topography, and thus the morphological characteristics of the anatomical structures. The relationships between the two can be visualized and even be compared with CT and MRI images [19,20]. Plastinated preparations have the disadvantage that over time, they lose their natural color (i.e., they turn yellow). This is influenced by the lipid content of the anatomical structures, and so removing the lipids with acetone and methyl chloride before immersion in the epoxy resin is an essential step in this technique [19,21]. However, the main disadvantages are represented by the need for specialized machines that allow for a long duration of vacuuming under a low temperature, as the process can take several weeks to several months to complete. It is thus clear that the use of any type of resin to coat the anatomical specimens needs to render the tissue hydrophobic in order to best be impregnated by the resin, and air or incomplete dehydration lead to the formation of bubbles or render the tissue cloudy and non-transparent.

Faster and more simple protocols have also been proposed that protect the tissue against the hydrophobicity and the oxidizing effects of the activated resins, either by first embedding the tissue in increasing concentrations of glycerin or first dehydrating the specimen in increasing ethanol solutions, then embedding it in glycerin [6,22]. Then, the specimen embedded in glycerin is finally cast in activated resin in custom-designed molds of plexiglas plates sealed with silicon gaskets. The main advantage of this technique is that the protocol is fast, and the final specimens can be easily sanded and polished. The protocol depends, however, on the resin utilized, and placing the specimen into the desired position between successive layers of cured resin is not an easy task. Chisholm and Varsou (2018) have shown that using the layer-by-layer casting technique is compatible with very large anatomic specimens such as full-body sections or entire limbs [23]. This same paper shows that imbalanced resin/activator ratios can lead to fast exothermic reactions that can boil the tissue and produce bubbles at the interface between tissue and resin, not to mention that fast polymerization is sometimes accompanied by the formation of toxic fumes [23]. Our current protocol is faster, although with lower polymerization times; it produces no toxic fumes; vacuum ensures the removal of all bubbles and a much better tissue impregnation with the resin/glycerol; and the positioning of the sample can be performed simply with plexiglas supports that are included within the final cured block.

Here, we have aimed to optimize a simple protocol and procedure for plastic resin embedding of full anatomical organs or sections, without a clearing step in order to preserve fats, and with a simple and reproducible casting technique that maintains the desired position of the sample in the resin block. Therefore, we have chosen three commercially available resins that are commonly used for modeling techniques and are widely available in hobby stores, and we tested them for fast tissue embedding, either after ethanol dehydration or after dehydration and glycerin pre-embedding. Moreover, we have developed a number of improvements to the existing methodology by utilizing cheap polypropylene food storage containers that are perfectly compatible as detachable molds for all tested resins, allowing for a better and undistorted view of the display specimens due to their cylindrical geometry when compared with parallelepipedal blocks of resins. Further, we utilized transparent acrylic rods to maintain the specimens in the desired positions, eliminating the need for multiple-layer casting. In fact, all our resins had densities above that of the tissue, and all the specimens were initially floating; therefore, even multi-layered casting would not have solved the problem of positioning the specimen. Acrylic rods were left in place, and since they had almost the same refractive index as the resins, they were only barely visible on the polished surface of the final resin blocks. The protocol does not use any solvent, and the option to use a pre-embedding glycerol step exposes the tissue either more or less to the oxidizing properties of the resin. Thus, the color of the final specimen can be controlled, allowing for more or less contrast depending on the purpose of the display sample. The fact that the protocol does not use clearing agents can be further expanded upon by utilizing some histological stainings to better visualize macroscopic structures, such as for, example, the adaptation of a Luxol fast Blue staining for visualizing myelin on whole brain slices, or the injection of different dyes into the vasculature or other canalicular structures in order to make them more clearly visible.

Despite fixation, bacteria and molds might grow in specimens over time. Therefore, we have also included UVC exposure times before and after embedding, as the transparency of the resins to UV light could not be determined from the products’ data sheets [24]. Exposure to UVC light also revealed that two of the resins did not show any yellowing effect after a 5 days of continuous exposure. A number of limitations still exist, such as the need to dehydrate the tissue and the need to use a vacuum pump/chamber and UVC light. However, the former equipment is readily available in hobby stores, and the latter is available in virtually all institutions in the form of sterilization lamps after the COVID-19 pandemic.

## 5. Conclusions

Altogether, we have presented a simplified, comprehensive, thoroughly descriptive, step-by-step protocol for performing whole-organ plastic embedding with a minimum amount of equipment and with commercially available supplies for forensic/anatomical, teaching, and research purposes. This proof-of-principle work shows that the protocol can be applied for most types of tissue, with promising possibilities for preparing large specimen volumes.

## Figures and Tables

**Figure 1 biomedicines-11-01433-f001:**
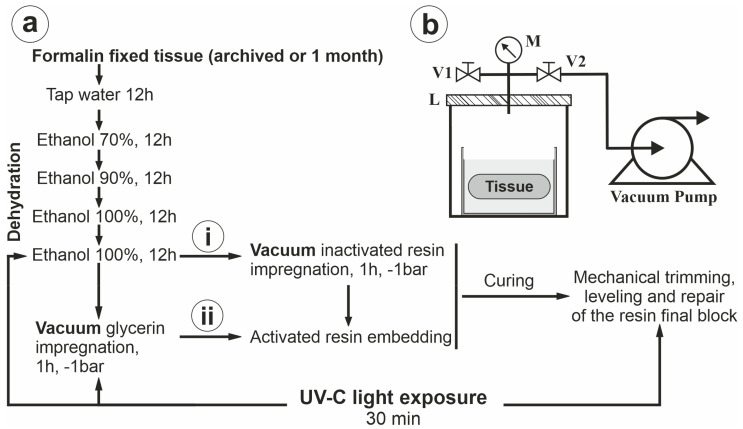
Proposed working protocol. (**a**) Steps and times are given for tissue that has been fixed routinely in 10% neutral-buffered formalin for 1 month or more. These involve washing the excess fixative solution, gradual dehydration, and processing for either (i) direct resin vacuum-forced impregnation or (ii) first glycerin vacuum-forced impregnation and then resin embedding without vacuum, in order to maintain a layer of glycerin as the interface between resin and tissue; after curing, the resin block can be trimmed and sanded. Moreover, blocks can be exposed to UVC light before and after resin casting in order to sterilize the tissue and protect against future growth of bacteria and fungi. (**b**) Schematic diagram of vacuum chamber with transparent acrylic lid (L), pump, valves (V1 and V2), and pressure gauge (M).

**Figure 2 biomedicines-11-01433-f002:**
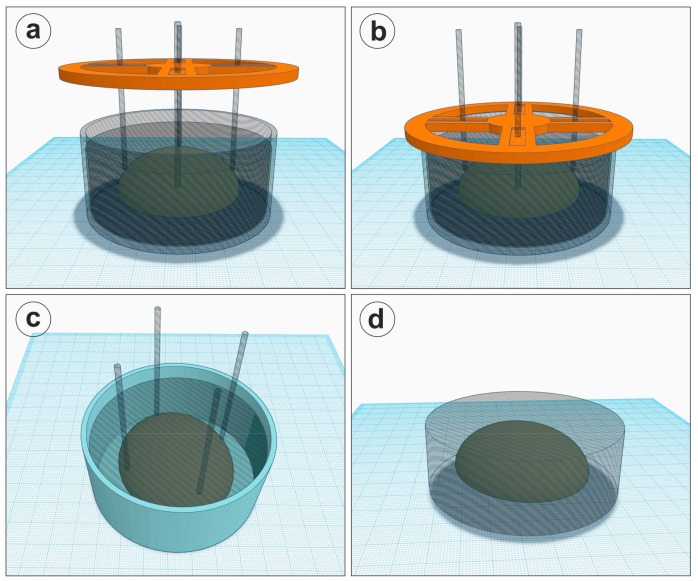
Schematic diagram showing the positioning of the tissue block during resin polymerization. (**a**) The activated resin is poured into a cylindrical jar of desired size, and the organ is positioned using 4 transparent acrylic rods that will be maintained at the desired depth utilizing a 3D-printed lid (orange). (**b**) The lid is then lowered until it can sit on the edge of the jar, and its weight keeps the tissue submerged in the desired position. (**c**) After the resin is polymerized, the lid is removed with the free acrylic rods included in the resin block. (**d**) The resin block is then removed from its mold, the rods are cut, the upper surface is sanded, and since the refraction index of the resin is the same as for acrylic plastic, the rods will hardly be detectable in the block and will not affect the visualization of the sample. Drawings were created in Autodesk Tinkercad (Autodesk Inc., San Francisco, CA, USA).

**Figure 3 biomedicines-11-01433-f003:**
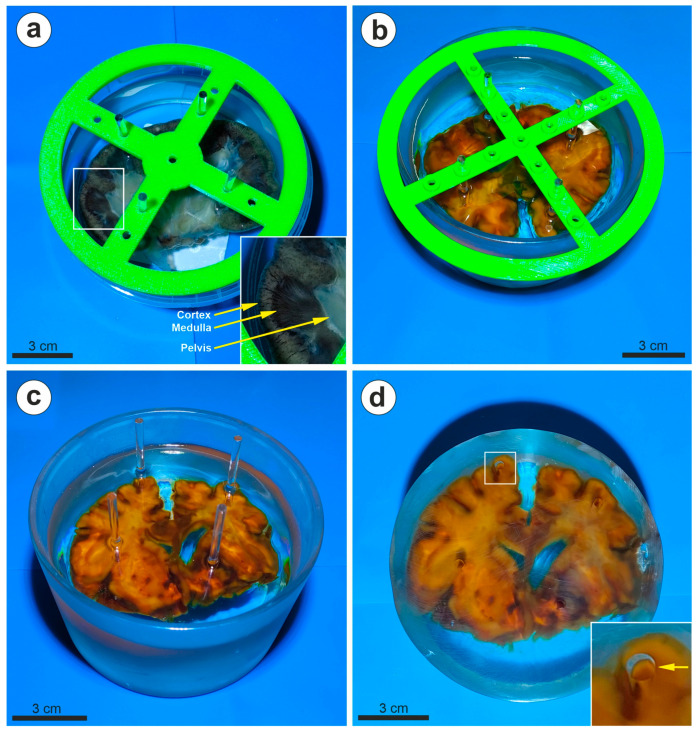
Examples using 4 mm plexiglas rods and 3D-printed support during polymerization. (**a**) Frontal kidney section (archived tissue) processed for inclusion with glycerin pre-embedding, being kept submerged in the resin by the plexiglas rods and the 3D-printed lid; the cortex and the medullary of the kidney are perfectly discernable after resin polymerization (inset image represents the enlarged detail). (**b**) The same process is exemplified here on a frontal brain slice (archived tissue, no glycerin pre-embedding). (**c**) After polymerization, the 3D-printed lid is removed, and the rods can be seen together with the wavy upper surface of the block. (**d**) After removal of the jar and leveling of the upper surface, the rods are barely visible (inset image and arrow). Here, we tested a glass jar, and the glass was extremely difficult to remove from the resin. A lot of polishing was thus necessary—using polypropylene molds and acrylic rods is the best choice.

**Figure 4 biomedicines-11-01433-f004:**
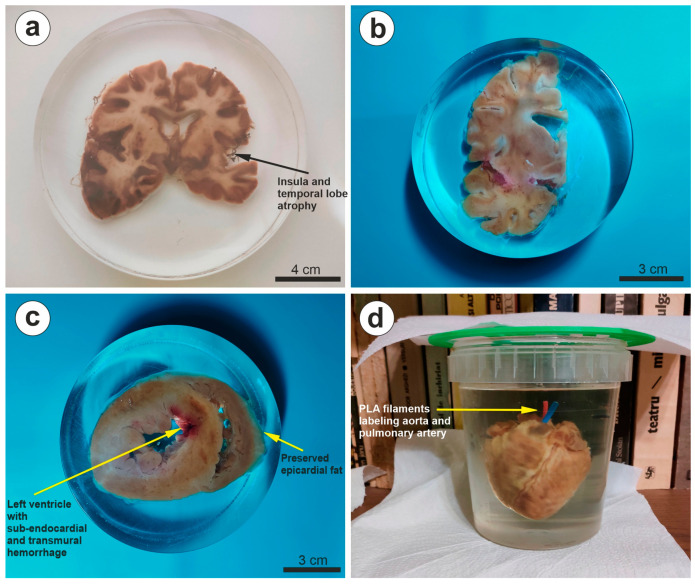
Examples of final results and use of colored PLA indicators. (**a**) A frontal brain section (1 month fixation) is presented here in a large cylindrical cast, illustrating that increased gray/white matter contrast is obtained without glycerin processing, as opposed to (**b**) a glycerin-processed hemisphere frontal section (1 month fixation), in which the contrast is much diminished. (**c**) A horizontal heart slice (1 month fixation) offers perfect transparency all around the organ, and (**d**) a whole heart is shown here (archived tissue) being supported by the acrylic rods and their 3D-printed lid (green) in the polypropylene mold, with red and blue PLA filaments to label vessels at the base of the organ. From this lateral view, the acrylic rods are completely invisible in the resin. The non-fuming block was kept on a shelf in our library until it was completely cured.

## Data Availability

All raw images are available from the authors upon request.

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
