# Peer review of "Simple Universal Whole-Organ Resin-Embedding Protocol for Display of Anatomical Structures"

_biomedicines, 2023, doi:10.3390/biomedicines11051433_

Round 1
Reviewer 1 Report
In the study of anatomy, a major problem is the sufficiency of fresh and fixed preparations. This paper describes a method for preserving internal organs. The material is sufficient. The authors should include photographs of all the organs performed and not only of the brain and heart. it should also be specified which section of the intestine was preserved, the same applies to the lungs. References are adequate.
[76] which section of the intestine?
[76] has the intestine been previously flushed?
[76] entire left and right lungs or some isolated lobes?
Author Response
In the study of anatomy, a major problem is the sufficiency of fresh and fixed preparations. This paper describes a method for preserving internal organs. The material is sufficient. The authors should include photographs of all the organs performed and not only of the brain and heart. it should also be specified which section of the intestine was preserved, the same applies to the lungs. References are adequate.
Q: [76] which section of the intestine?
[76] has the intestine been previously flushed?
[76] entire left and right lungs or some isolated lobes?
Re: We thank the Reviewer for the appreciations, and considering the points raised by her/his questions, we have included a new figure in the manuscript (Supplementary Figure S3) which illustrates lung, large intestine and spleen section blocks. Moreover, we have clarified in the manuscript that we tested here only large intestine fragments dissected and prepared together with a part of the mesentery, slices of isolated lung lobes, and that all organs were thoroughly washed before the inclusion protocol (intestine was flushed of its content, and the lung was cleared of any visible thrombi that might precipitate during inclusion). As this work is also expanding into trials to pre-stain the tissues prior to inclusion, we have also included here an example of a lung injected with a red dye into a segmental bronchus, thus illustrating the sharp edge between functionally independent lobules (as stained/unstained parenchyma). The methodology of dye injection has been described in the Materials and Method section.
Reviewer 2 Report
State the gap of study. Chisholm and Varsou (2018) published an article related to resin‐embedded transverse cross‐sections. What is the strength of your study / the improvements that have been made since the former study?
State the ethic clearance for the study.
In need of proofreading. Choose the proper verb tense and carefully construct the sentence when writing up a scientific paper. eg Line 85-87, line 98-99.
To show a more convincing images of "final results" with proper labelling than the ones in Figure 3 and Figure 4. Have you achieved the objectives?
-
Author Response
Q1: State the gap of study. Chisholm and Varsou (2018) published an article related to resin‐embedded transverse cross‐sections. What is the strength of your study / the improvements that have been made since the former study?
Re1: This suggested paper has been included in our manuscript as reference 23, as these authors showed images exemplifying problems and issues that may occur during inclusion, such as the fast-exothermic reaction that can produce gases and destroy the sample (an issue that we have also encountered and reported). As the Reviewer has suggested, we have described in more details, in our updated manuscript, the advantages/ disadvantages of our method compared to this paper. The main advantage of this paper, compared to our study, is that they had available full donor bodies solely for this purpose, while our pilot study started from available fixed (fresh or archived) dissected organs from teaching cadavers. Our largest casted blocks contain only 700ml of resin [the brain sections and large intestine samples (an intestine image has been now included in the updated manuscript in the Supplementary Figure S3], the PlexiFluid 2 resin has no upper limit regarding the working volumes, and this keeps a door opened for anything to be included (we have also underlined this in the revised version of the manuscript). Although we did choose the largest vacuum chamber available, we are also limited for now by its size (19l, diameter=28cm, height = 30cm). This paper is the result of a small project implemented in our anatomy/histology departments and it is the proof-of principle that a small budget can also lead to good transparent blocks of tissue. Thus, this work will continue and we are aiming at including also whole organs in the future (a whole lung can fit on our vacuum chamber already), and a custom-made wider vacuum chamber might also be produced.
Our protocol, has, however, many improvements over the above report, such as: (i) a much faster total protocol (true that for smaller blocks for the moment), (ii) the use of non-toxic and non-fuming commercially available resins with much slower polymerization rates and less exergonic reactions that lead to virtually no boiling and bubbling of the tissues, (iii) the use of vacuum, which is ideal to remove the bubbles that form inherently when mixing a large quantity of resin mixture, and is also essential to force-impregnate the tissue with a containing solute (un-activated resin or glycerol in our case, to prevent drying and bubble formation in time), (iv) the use of UVC light to prevent the growth of bacteria and molds in the post-fixed tissue, that might again affect it in time, and (v) the simplicity of the protocol itself. Thus, we did not cast the volumes layer-by layer in order to trap our tissue slices as “floating” inside, but we instead used plexiglas supports that maintained the tissue at the desire depth inside the block, and in the end, these supports were hardly visible in the finite block, having the same refraction index. Casting multiple layers might result in a visible stratification of the final block (even small deviations in the activator concentration will lead to different curing speeds), and thus casting the block in one go also improves the lateral view of the specimens (our samples are not only flat disks, but also taller cylinders with a clear view on the lateral sides of the specimens (Please see Figure 4d and Supplementary Figure S3d).
Taking the suggestion of the Reviewer, we have summarized these differences in our Discussion section, in the updated manuscript, making it thus clearer, and for these improvements we thank the Refreee.
Q2: State the ethic clearance for the study.
Re2: The paper has the format recommended by the Editors of the Journal, and under the “Institutional Review Board Statement” we have described that we utilized available teaching material from our department. The study was approved by the Ethics Committee of our university when it began.
Q3: In need of proofreading. Choose the proper verb tense and carefully construct the sentence when writing up a scientific paper. eg Line 85-87, line 98-99.
Re3: We thank the Reviewer for her/his thoughtful evaluation. As recommended, we have carefully proofread our resubmitted manuscript.
Q4: To show a more convincing images of "final results" with proper labelling than the ones in Figure 3 and Figure 4. Have you achieved the objectives?
Re4: Both Reviewers requested more images, therefore we have added a new image composite showing other organ fragments processed following this protocol (Supplementary Figure S3). Scalebars and further labeling have been included in all images when needed. We hope the Reviewer will find that these additions improve the strength of our manuscript and support its results. Considering that our objective here was to show the proof-of-principle of a fast, low budget, but high-quality display samples preparing protocol, we did achieve this milestone; of course this work is just in the beginning, and many samples (stained and unstained) will be needed. In the same endeavor, we have launched our high-resolution virtual histology database (whole slides scanned as 80x magnification) [http://86.122.148.72/DSServer/Login (name/password: student)], and we intend to link these slides with casted organ blocks through QR codes for teaching purposes.